# “Salt and Eat It or No Salt and Trash It?” Shifts in Support for School Meal Program Flexibilities in Public Comments

**DOI:** 10.3390/nu17050839

**Published:** 2025-02-28

**Authors:** Sarah Moreland-Russell, Natasha Zimmermann, Jessica Gannon, Dan Ferris, Charles Alba, Rebekah R. Jacob

**Affiliations:** 1Prevention Research Center, Brown School, Washington University in St. Louis, St. Louis, MO 63130, USA; jessica.gannon@wustl.edu (J.G.); rebekahjacob@wustl.edu (R.R.J.); 2School of Public Health, Yale, New Haven, CT 06510, USA; n.zimmermann@wustl.edu; 3Center for Social Development, Brown School, Washington University in St. Louis, St. Louis, MO 63130, USA; dan.ferris@wustl.edu; 4Brown School, Washington University in St. Louis, St. Louis, MO 63130, USA; alba@wustl.edu

**Keywords:** National School Lunch Program: National School Breakfast Program, school meals, policy, Federal Register, United States Department of Agriculture, meal flexibilities, federal food program legislation

## Abstract

**Background:** The Healthy, Hunger-Free Kids Act was passed in 2010 to update nutrition standards in the National School Lunch and Breakfast Programs to be in accordance with evidence-based guidelines. In 2017 and 2020, the United States Department of Agriculture proposed flexibilities to the nutrition standards for milk, whole grains, and sodium. **Objective:** This study examines the positions used by stakeholders in support for or opposition to the proposed rules. **Methods:** We conducted a longitudinal qualitative content analysis of public comments posted to the U.S. Federal Register in response to the USDA’s proposed rules in 2017 and 2020. All public comments submitted by individuals and organizations were analyzed (n = 7323, 2017 and n = 2513, 2020). **Results**: Results indicated three categories of arguments: (1) comments favoring the original law, (2) comments favoring flexibilities, and (3) other. In both comment periods, constituents opposed the implementation of flexibilities, while schools and manufacturers/industry predominately supported them. Academic and advocacy organizations opposed the original proposed change (2017) but relaxed their position in 2020. **Conclusions**: Any flexibility to the required nutritional standards of school meals has the potential to affect the health trajectory of youth. It is imperative to understand how stakeholders view this issue and inform policy change.

## 1. Introduction

Obesity among people aged 2–19 in the U.S. is a profound issue that affects an increasing number of youth [1]. Specifically, for this age range, obesity (body mass index values at or above the 95th percentile of the growth charts) has increased from 13.9% from 2000 to 19.3% in 2018, and severe obesity (body mass index at or above 120% of the 95th percentile) increased from 3.6% to 6.1% [1]. Childhood obesity leads to the development of a variety of health conditions in childhood, adolescence, and adulthood, including type II diabetes, cancer, coronary heart disease, cardiovascular disease, and metabolic disease [2]. Furthermore, obesity disproportionately effects Black, Hispanic, and youth from households with low incomes [3,4,5].

The National School Lunch and School Breakfast Program (NSLP/SBP) in the U.S. are government initiatives aimed at providing nutritious meals to students during the school day. These programs serve as a crucial component of ensuring that children have access to balanced meals, which can positively affect their health, academic performance, and overall well-being [6]. The U.S. Department of Agriculture (USDA) administers the School Lunch and Breakfast Programs. Any public school, nonprofit private school, or residential childcare institution can participate in the program and receive federal funds for each meal served, as long as they adhere to federal nutrition requirements set forth by the NSLP/SBP. The vast majority of schools participate in the program and, on a typical day, nearly 30 million children participate in the National School Lunch Program and 15.5 million children participate in the School Breakfast Program [6]. The programs provide participating schools with cash subsidies and food from USDA for the meals served. Schools that participate in the program must offer free or reduced-price lunches to eligible students, based on their family’s income level. The benefits of free and reduced-price options extend beyond providing nutritious meals. They increase access to food during the school day, regardless of their family’s financial circumstances [6].

Over time, the U.S. Congress and the USDA have advanced legislation and regulations that have affected school meal nutritional standards. Two landmark updates included the 2010 Healthy, Hungry-Free Kids Act (HHFKA) and the USDA’s 2012 final rule, *Nutrition Standards in the National School Lunch and School Breakfast Programs* [7]. These updates called upon the USDA to structure school meal program requirements to align with the practices recommended by the Food and Nutrition Board of the National Research Council of the National Academies of Science [7].

The new HHFKA regulations came into effect in the 2012–2013 school year [7]. The improvements aimed to increase the availability of fruits, vegetables, whole grains, and fat-free milk; reduce the levels of sodium; and meet the nutrition needs of school children within calorie requirements [7,8]. Specifically, schools were required to gradually (over 10 years) reduce sodium levels in school meals, offer only fat-free plain and flavored milk and low-fat plain milk, and 100 percent of grains offered had to be whole-grain-rich [7].

Research has demonstrated the effectiveness of this large-scale public health policy, documenting significant improvement in the nutritional quality of school meals [8,9,10,11,12,13]. Although implementing such broad-scale changes faced initial challenges, research shows that schools implemented the updated nutrition standards and began to offer healthier meals with more offerings of fruit and vegetables, significant reductions in sodium and the percentage of calories from saturated fat, and increases in fiber [8,9,10,11,12,13]. In addition, implementation of the HHFKA has had an effect on obesity in youth [14,15]. Kenney et al. [14] tested whether the HHFKA was associated with reductions in child obesity risk over time, and while they found no significant association between HHFKA and overall childhood obesity trends, for children in poverty, the risk of obesity declined substantially each year after implementation. Kenney et al. [14] also reported that obesity prevalence would have been 47 percent higher in 2018 if the HHFKA had not been implemented.

Despite documented success of the HHFKA, several changes have been adopted since its inception. Changes to the *Nutrition Standards in the National School Lunch and School Breakfast Programs* were proposed in 2017 with an interim final rule (82 FR 56703) and subsequent 2018 Final Rule (83 FR 63775), named the *Child Nutrition Programs: Flexibilities for Milk, Whole Grains, and Sodium Requirements* [16]. This rule allowed schools flexibility in implementing milk, sodium, and whole grain nutritional requirements during the 2018–2019 school year (SY 2018–2019). Specifically, schools were again allowed to offer flavored, low-fat milk; provide only half of the weekly grains as whole-grain-rich; and retain higher sodium levels [17].

In 2020, the Center for Science in the Public Interest sued the Department of Agriculture (*Center for Science in the Public Interest* et al. v. *Sonny Perdue, Secretary, et al.*) [18]. The U.S. District Court decided that the rule should be vacated because of a procedural error with the promulgation of the 2018 Final Rule (83 FR 63775) [19]. In response, the USDA proposed another rule, “Restoration of Milk, Whole Grains, and Sodium Flexibilities” Docket FNS-2020-0038 (85 FR 75241) [20], moving to make the aforementioned flexibilities permanent. The final rule went into effect in February 2022. The new standards took effect on 1 July 2022 and applied to the 2022–2023 and 2023–2024 school years. The changes were different from the previous flexibilities that were implemented and challenged but were still a departure from the original HHFKA requirements. For instance, instead of 50 percent of the required whole grains, the new standards required 80 percent. In addition, the milk requirement allowed for flavored, low-fat milk but at a cost to the student as a competitive beverage [21].

A key step in obtaining public opinion on new federal regulations entails publishing proposed rules in the Federal Register, a daily publication. The Federal Register is subdivided into four ordered sections of (1) presidential documents, (2) rules, (3) proposed rules, and (4) notices. Particularly relevant to researchers is the process established by the Administrative Procedure Act, which includes a structure for garnering public comments on regulation [22]. These comments, useful for understanding of public opinion regarding regulations, have previously been used to inform research studies, including ones related to USDA and its policies. For instance, Rose et al. (2021) analyzed public opinion via Federal Register comments about the USDA’s plans for labeling genetically modified foods [23] and Tobacyk et al. (2022) uncovered new health-based reasons for kratom use in the U.S. using comments published [24]. In addition, Haynes-Maslow et al. (2018) performed a thematic content analysis of public Federal Register comments about the USDA’s Supplemental Nutrition Assistance Program Retailer Rule by coding themes from a random sample of the comments [25].

The purpose of this study is to assess public opinion on the flexibilities to the National School Lunch and Breakfast Programs proposed and implemented by the USDA. While a vast amount of research has documented the original acceptance and implementation of the HHFKA [8,9,10,11,12,13], little research has been completed on the changes (specifically the flexibilities adopted in 2017 and 2020) imposed by the USDA that have been implemented since its original adoption. Our research questions include the following:(1)Was the predominant argument in favor or in opposition of the flexibilities?(2)How did support vary among different stakeholder groups?(3)What were the main topics used in comments in support and in opposition of flexibilities?

We analyze public comments expanding on the approach used by Haynes Maslow in the coding and thematic analysis of comments related to proposed rules FNS-2017-0021 (82 FR 56703) [7] published on 30 November 2017 and FNS-2020-0038 (85 FR 75241) [20] published on 25 November 2020 posted to the Federal Register. These meal flexibilities marked important changes for the NSLP and SBP, and their significance is further compounded by the profound effect that the NSLP and SBP have on the daily nutritional intake of children and adolescents.

This study examines comments posted to the Federal Register to assess support (or lack thereof) for the 2017 and 2020 proposed rules. We also analyzed other ideas expressed through the comments and compared results to understand the perspectives of the various stakeholder types involved.

## 2. Materials and Methods

### 2.1. Extracting Public Comments

To download all public comments for analysis, we requested an Application Programming Interface (API) key from Regulations.gov in July 2019. With access to the API, we used an open source Python (2019) web scraper tool [26] to retrieve all public comment text from the Docket (FNS-2017-0021) with accompanying data into a .csv file. The file was then cleaned to remove true duplicates (same individual/organization and same comment text), and identify and code duplicate and similar text (same or close to same form letter submitted by multiple individuals/organizations). Cleaning was completed in R [27] and the resulting clean dataset was written as an .xlsx file. We repeated the same process in June 2021 with Docket FNS-2020-0038 using a newly developed web scraper tool [28] due to major updates to Regulations.gov in 2020.

### 2.2. Qualitative Analyses

Comments were coded using a deductive qualitative approach for sentiment by two members of the research team in two separate time periods for each set of comments (2019 for 2017 comments and 2021 for 2020 comments). For both times periods, one team member coded all comments. A quality check was performed by another team member who randomly selected and coded 20% of the comments. The inter-rater reliability of 20% of the double-coded comments for the 2017 dataset was 0.96 and that for the 2020 dataset was 1.00. For 2017 comments, coders identified if each comment’s main stance was to (1) implement newly proposed flexibilities, (2) maintain current regulations, (3) implement some of the new flexibilities, (4) implement a different option (new idea), (5) comment on general nutrition, or (6) something else (other). For 2020 comments, because the proposed rule regarded restoring the original flexibilities, the research team coded each comment as follows: (1) if the commenter did not want to restore flexibilities, (2) wanted to restore all flexibilities, (3) wanted to restore some of the flexibilities, (4) implement a new option (new idea), (5) commenting on general nutrition, or (6) something else (other). The other category usually contained random comments often not relating to the flexibilities at all and accounted for a less than 1% of comments. For example, one comment coded as “other”: *“These are the children who will ultimately be responsible for our care in old age”*.

To examine support for flexibilities across the two time periods, we collapsed categories into the following codes: (1) *In favor of original HHFKA*—maintain current regulations (2017) comments or do not restore flexibilities (2020); (2) *In favor of flexibilities*—implement new rules/implement some of the new rules (2017) or restore all flexibilities/restore some of the flexibilities (2020); and (3) *Other*—implement another option, general nutrition, and other (2017 and 2020). No comments were excluded from the analyses.

Organization name was gathered from the original commenter submission (supplied by Regulations.gov) and was added if identified specifically within the comment text. We categorized organizations into the following: academic, advocacy, constituent (no organization specified, only first and last name provided), manufacturer/food industry, school, other governmental agency (non-school), and other. Two coders categorized the organization based on the organization name and, for those that were not obvious, we searched for the organization’s webpage to determine its primary representation.

We calculated descriptive statistics to assess the support variable across the two time periods and across organization type. Comment coding was completed in Microsoft Excel and data management and analyses were performed using R [27]. All coded comments and analysis files for this manuscript can be found at https://github.com/rebekahjacob/federal-register-comments-school-lunch.

Finally, we used topic modeling to identify recurring topics behind these massive amounts of textual data. Specifically, we used BERTopic [29], represented across distinct organizations by normalized comment frequency, to visualize topics most present in the submitted comments spanning 2017 and 2020 across organizations. BERTopic is a topic modeling technique that uses a pre-trained language model to group contextually similar comments into clusters and applies a statistical distribution to represent each cluster with its most prominent keywords. We chose BERTopic because it leverages pre-trained language models, which allows it to better semantically contextualize these comments and generate more meaningful topics compared to traditional topic models like LDA. Separate BERTopic analyses were performed among those supporting flexibilities and in favor of the original HHFKA due to the expected semantic differences amongst their comments. Each defined topic string is presented by the keywords.

## 3. Results

A total of 10,441 comments were published for the 2017 and 2020 rules (7523 and 2918, respectively). Form letters written by one main group (usually an advocacy organization) and submitted as public comments by other entities, comprised 5646 (75.0%) of the comments posted in 2017 and 2452 (97.6% of total) of the comments posted in 2020. Some of the advocacy organizations most represented in form letters were American Heart Association, Salud America!, and the School Nutrition Association. Table 1 outlines the categories that were coded and provides examples of the types of comments that fell into each category.

### 3.1. Organization Representation

Table 2 highlights the percent of comments submitted by organization type for each of the two rules proposed (2017, 2020). Only a small portion of the comments listed an organization. Those without a listed organization we coded as a constituent. Public comments for both proposed rules were most often submitted by constituents. General constituent comments were submitted mostly using form letters provided by advocacy organizations including Salud America! and American Heart Association. In 2020, when the rule was focused on reinstating the flexibilities, schools submitted a greater number of comments when compared to their comment activity in 2017. The School Nutrition Association form letter was often used by schools who posted comments.

### 3.2. Stance in 2017 Versus 2020 by Different Organizations

Figure 1 compares the dominant organizational stance in 2017 and 2020. In 2017, when stakeholders were responding to the initial proposed implementation of the flexibility rules, the manufacturers/industry organizations and the school stakeholders were the most in favor of the milk, whole grain, and sodium flexibilities. Many of the comments by schools focused on supporting flexibilities so that more students would eat the food provided.

*We’ve changed products, reduced sauces, reduced condiments, don’t cook with or serve salt, etc. We still are barely hitting the accepted sodium level. Food needs taste in order for kids to eat it. So the debate- salt and eat it or no salt and trash it leads to hunger and disrupted classrooms. I’d like to add that I’m a Registered Dietitian and SNS certified. I believe in the NSLP. I think there’s a bit too many regulations to make great menus. I think the regulations are confusing to staff and the public. We as directors then have to pick up the pieces”*.(posted by school organization, 2017)

Manufacturer and industry stakeholders posted comments specific to the type of product they provided. For instance, the National Dairy Council posted comments supporting the flexibility for allowing flavored milk:

*“Milk served in the school meal programs is required by law to be ‘consistent with the most recent Dietary Guidelines for Americans.’ The 2015–2020 DGA recommends low-fat or fat-free milk. Both low-fat and fat-free flavored milk could be considered ‘consistent’ with the DGA, meeting the legal requirement. The law permits ‘flavored and unflavored’ milk in schools. The 2015–2020 DGA recognizes that ‘some sweetened milk and yogurt products may be included in a healthy eating pattern as long as the total amount of added sugars consumed does not exceed the limit for added sugars, and the eating pattern does not exceed calorie limits’”*.(posted by a manufacturer, 2017)

On the other hand, academic organizations, constituents, and other government agencies (non-school) submitted comments that were predominantly in favor of the NSLP and SBP as originally defined by the HHFKA, with only 0–2% of comments by these stakeholder types in support of the flexibilities.

*“I am a researcher and for the health of Latino and all children, I urge the USDA to maintain strong nutrition standards for meals served in schools. Providing flexibility by allowing schools to serve grains that are not whole-grain-rich and flavored milk with 1% fat would constitute a direct contradiction of the current dietary guidelines and result in a step back on the progress already made in promoting a healthier lifestyle, a healthy weight, and overall health equity. About 1 in 3 Latino families live in poverty, and 1 in 4 are food insecure, according to a 2017 Salud America! research review. For many Latino kids, school is their only chance to get a well-balanced meal. Thats why I urge you now to keep the bar high when it comes to serving nutritious food in schools across America and not reduce the nutritional quality of school food”*.(posted by academic organization, 2017)

In 2020, after the flexibilities were allowed and implemented in several schools, stakeholder perspectives remained clear among some stakeholders, but were less delineated among others. School and manufacturer stakeholders did not stray from their original position, strongly supporting (and even more so) the continued implementation of the flexibilities. Constituents also stuck to their original opposition to the flexibilities and commented on keeping the original HHFKA in place. Academics and other non-governmental agencies (NGOs), however, became less decisive. Both groups posted slightly more comments in support of flexibilities (academics n = 24, NGOs n = 11) compared to the original 2017 comments (academics n = 18, NGOs n = 8). Part of this change may be a reflection of changes in participation or the need to implement flexibilities in response to the challenges to food supply borne by the pandemic. For example, a common sentiment posted related to the need to adapt based on the pandemic:


*“Covid 19 has caused a huge disruption in our school lunch and breakfast programs and the kids and staff will need an adjustment time to allow a degree of normal to return. We will need that time to increase our participation back to normal levels before adding more restrictions.”*
(posted by school, 2020)

### 3.3. Opinions on Types of Flexibilities to Restore

Many of the comments posted in both 2017 (n = 50) and 2020 (n = 55) supported implementing only some (and not all) of the flexibilities.

In 2017, the type of flexibility most supported among those who voiced support for only some of the flexibilities was retaining higher sodium levels (n = 20, 40%). Twenty-eight percent supported keeping the whole grain flexibility, which allowed for only half (instead of 100%) of the weekly grains to be whole grain. Only 8% (n = 4) posted a comment that only supported the milk flexibility, which allowed schools to offer flavored, low-fat milk. Schools, constituents, and manufacturers primarily posted these comments in 2017.

In 2020, the two types of flexibilities most supported among those who voiced support for only some (and not all) of the flexibilities were the milk (n = 17, 31%) and sodium flexibilities (n=7, 31%). Twenty-two percent (n = 14) supported keeping the whole grain flexibility. Again, manufacturers, schools, and advocacy groups primarily posted comments supporting implementation of some of the flexibilities. However, unlike 2017, academic organizations also posted comments supporting the implementation of some of the flexibilities in 2020.

### 3.4. Other Topics Present in Comments

Figure 2 and Figure 3 compare the main topics found in comments submitted by organizations in support of the flexibilities (Figure 2) and those in support of maintaining the original HHFKA requirements (Figure 3) in 2017 and 2020.

For those supporting flexibilities, topics present in comments were generally consistent within manufacturer, school, and advocacy organizations in 2017 and 2020: manufacturers, schools, and advocacy groups submitted comments in both years related to “flavored, nutrition, flexibility” (topic 0). Academic organizations and other government agencies did not support flexibilities in 2017, as indicated in Figure 1, but commented about “students, rules, need” (topic −1) and “flavored, nutrition, flexibility” (topic 0) in 2020. On the other hand, topics in the constituent group comments mostly varied from year to year: in 2017, the majority of the constituents commented about “students, rules, need” (topic −1), but in 2020 the majority of constituents shifted their comments primarily toward “flavored, nutrition, flexibility” (topic 0).

For those in favor of keeping the original HHFKA there was a noticeable semantic shift and greater variability among submitted comment topics within groups in 2017 compared to 2020. For example, academics in 2017 mostly discussed “children, nutrition, need” (topic 0) but this shifted to “children, standards, nutrition, usda” (topic −1) in 2020. Similarly, advocacy groups primarily commented on “nutrition, flavored, usda,” (topic 4) in 2017 but shifted their focus toward “children, standards, nutrition, usda” (topic −1) in 2020. Additionally, among constituents, in 2017 they heavily commented on “children, nutrition, need” (topic 0) but this shifted to “meals, disparities” (topic 3), in 2020.

## 4. Discussion

Nearly 30 million children receive lunch and 15.5 million children receive breakfast on a given day from the National School Lunch and Breakfast Programs. The nutritional standards of the NSLP and SBP therefore have significant impacts on the dietary intake of children and adolescents, with implications for a variety of educational, behavioral, and health outcomes. It is therefore imperative to understand how individual constituents, school employees, manufacturers, and a variety of other stakeholders view (in support or opposition) changes to these programs.

This is the first longitudinal examination of comments posted to the U.S. Federal Register in response to proposed rule changes to the NSLP and SBP to allow for flexibilities in milk, whole grain, and sodium. Specifically, we examined comments made in support and opposition of the allowed implementation of the flexibilities among schools at two time points: in 2017 when the initial flexibilities were proposed, and in 2020 when the USDA proposed to restore the flexibilities after a challenge in federal court. In both 2017 and 2020 comment periods, constituents resoundingly opposed the implementation of flexibilities. On the other hand, school agencies and manufacturer/industry organizations predominately supported the implementation of all flexibilities in both comment periods. Academic and advocacy organizations and other non-school governmental agencies opposed the original proposed change (2017) but somewhat relaxed their position in 2020. Some of the posted comments supported implementing some (and not all) of the proposed flexibilities. Most of these comments were posted by schools who had been successful in implementing some of the original requirements but were experiencing supply issues (e.g., lack of adequate supply of whole grain products) or drops in participation. Manufacturers primarily commented only on the flexibilities that related to their specific product (i.e., dairy manufacturers only commented regarding the milk flexibility).

Regardless of year, there was a central argument that resounded among the groups. For schools and manufacturers, prioritizing participation and the likelihood of students eating the food provided prevailed over always providing healthier options that meet higher nutritional standards. Constituents focused on choosing the healthier option and maintaining the nutrition standards set forth by the original HHFKA legislation.

When reviewing topics present in comments submitted by organizations in support of flexibilities, we found comment topics to be generally consistent within manufacturer, school, and advocacy groups in both 2017 and 2020. Constituent comments covered a variety of topics and signaled a greater semantical shift in submitted comment topics between 2017 and 2020 compared to other stakeholder groups. For those organizations supporting the original HHFKA, there was much less consistency of comment topics within groups or across years (2017 and 2020). These findings suggest less alignment among groups who were in support of keeping the original HHFKA regulations and more alignment among groups who wanted the flexibilities. This is important when considering how to build support and facilitate effective policy development and implementation across distinct entities. These topic models could provide valuable insights towards implementing strategies when garnering support for such policy changes. For instance, among constituents in favor of keeping the original HHFKA, the shift from “Nutrition needs for children” (“children, nutrition, need” per topic 0 in Figure 3) to “meal disparities” (“meals, disparities” per topic 3 in Figure 3) suggests that a shift in focus to addressing disparities-related concerns when tailoring communications or making policy adjustments could garner more trust and possibly action among constituents to support these flexibilities.

Another interesting finding in this study is that most of the comments were submitted by constituents in both the 2017 and 2020 as form letters, and most of the form letters came from specific lobbying agencies—either supporting or opposing the flexibilities. Therefore, while specific organizations can only submit one comment to represent their stance, the use of form letters by constituents in alignment with their stance allows them to have a much greater numerical presence. Essentially, organizations with vast networks who can convince constituents to support their position and submit their form letter can change the balance of support for or against proposed rules. This can ultimately impact how the federal agency responds, as it must provide a summary of the comments and provide a response to the public comments to justify final decisions. While providing comments regarding proposed bills is an important responsibility for any constituent, it is important to understand the potential conformity bias that form letters may present.

In addition, this study is one of the few examinations of federal comments posted in response to a public health policy. In our review of the literature, we found only three [23,24,25] published research articles that evaluated comments in the Federal Register posted in regards to a public health policy. While the USDA and other executive branch agencies are responsible for reviewing every single comment and providing a summary of comments and mechanisms for response to comments, entities outside of the executive branch (specifically academia and other evaluative organizations) have not been able to easily access or examine the comments posted. The amount of work involved in trying to access raw data from the Federal Register is prohibitive and results in very little academic research examining Federal Register comments. The executive branch has recently added a new function (in 2022) for bulk data download [30], which will hopefully make the process more transparent and access to comments more readily accessible for further research and evaluation of comments posted [27].

Obesity in youth is already a widespread issue within the US [3]. While there is not yet an established causal relationship between school meal programs and obesity, the amount and type of food consumed at school is significant. Unfortunately, most youth in the U.S. do not meet the recommended servings of foods that provide for proper nutrition, optimal growth, and development, and consume too many of the foods that lead to poor health outcomes [9,31,32]. Poor nutrition already disparately effects Black, Hispanic, and American Indian/Alaskan Native children and children living in under-resourced house-holds, putting these populations at the greatest risk for obesity and diet-related disparities [5]. School food consumption can therefore be important in attaining proper nutrition. Ensuring that programs such as the NSLP and SBP are serving the best interests of communities or populations already suffering nutrition and health related inequities is a pressing task. It is therefore critical to understand how decisions across the many stakeholders involved in this system, and specifically by schools to implement these flexibilities, result in variations of food served food to children across the U.S. Further research is needed to understand these variations, specifically in terms of how they may result in health-related disparities.

### Study Limitations

The data for this study were collected from comments on the U.S. Federal Register so may not be representative of the larger public and stakeholder opinions. While the U.S. Federal Register is open to any person wishing to submit a comment regarding any proposed rule, some people may not have access to a computer, be aware when the federal comment system was open, or feel that submitting a comment could have an impact on the policy decision making by the federal agency. In addition, some districts may actively encourage parents or stakeholders to actively engage and submit comments on the U.S. Federal Register. As a result, comments across stakeholders from these districts may be overrepresented, potentially leading to an imbalance in the dataset. As a result, methods such as BERTopic may amplify these biases in representation. Furthermore, parents or educators that stem from an older demographic may be less active online, and thus may be under-represented in the dataset, as public comments are primarily collected from digital platforms. Consequently, stakeholders such as parents or educators from older demographics may have lower levels of online engagement, leading to their underrepresentation among such datasets collected from digital platforms. Additionally, active individuals on the Federal Register who are less directly affected by the policy may be influenced by previously submitted comments, leading them to echo popular sentiments that resonate with them [33]. This could result in the over-amplification of popular topics reflected in our topic models.

Moreover, the stances that were used in coding were relatively broad, so there could be variation within comments listed under the same stance, along with some comments falling in a gray area between stances. We attempted to mitigate this potential limitation by checking a sample of codes in each dataset (2017 and 2020) and testing inter-rater reliability.

## 5. Conclusions

Any flexibility to the required nutritional standards of school meals has the potential to affect the health trajectory of youth. It is imperative to understand how stakeholders view this issue and inform policy change.

## Figures and Tables

**Figure 1 nutrients-17-00839-f001:**
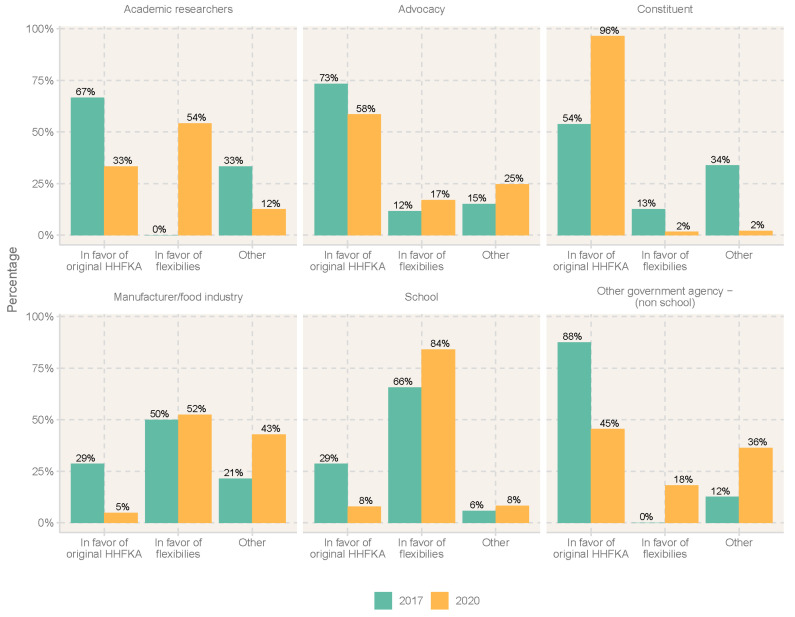
Stance by organization type in 2017 and 2020. Figure 1 compares organizational support of the 2017 (green bars) and 2020 (orange bars) proposed rules allowing for flexibilities for milk and whole grain to be implemented in US National School and Breakfast Programs. HHFKA stands for the Healthy, Hunger-Free Kids Act of 2010 federal legislation, which resulted in the 2012 final rule, *Nutrition Standards in the National School Lunch and School Breakfast Programs*.

**Figure 2 nutrients-17-00839-f002:**
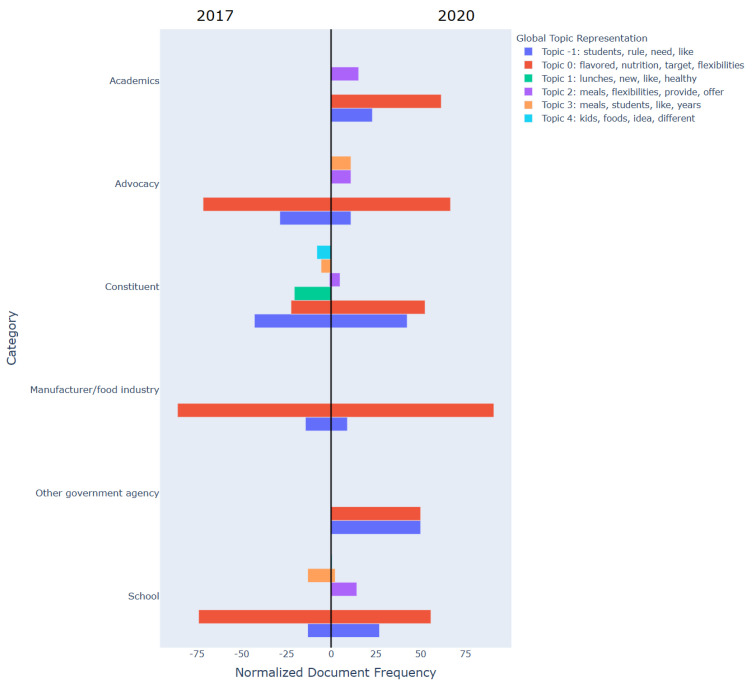
Comparison of topics present in comments among organizations in favor of flexibilities in 2017 and 2020. Figure 2 compares topics present in comments among organizations in support of flexibilities in 2017 and 2020. Topic strings are represented by color. The x-axis represents the topic models across each organization type whilst the y-axis represents the number of comments from an organization that have been assigned to a specific topic, normalized by the total number of comments from that organization $\left(\frac{{Count}_{topic,org}}{{Count}_{org}}\times 100\right)$.

**Figure 3 nutrients-17-00839-f003:**
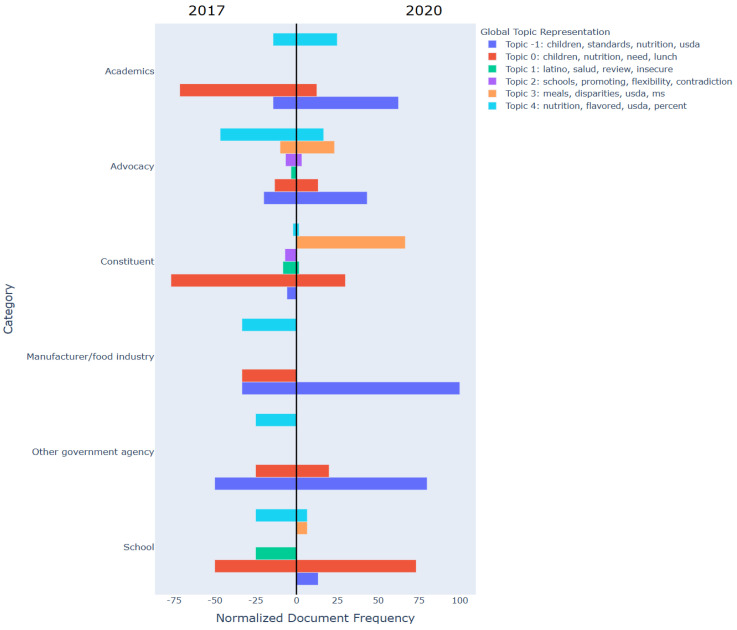
Comparison of topics present in comments among organizations in favor of keeping the original Health Hunger Free Kids Act regulations. Figure 3 compares topics present in comments among organizations in support of keeping the original Hunger Free Kids Act regulations in 2017 and 2020. Topic strings are represented by color. The x-axis represents the topic models across each organization type whilst the y-axis represents the number of comments from an organization that have been assigned to a specific topic, normalized by the total number of comments from that organization $\left(\frac{{Count}_{topic,org}}{{Count}_{org}}\times 100\right)$.

**Table 1 nutrients-17-00839-t001:** Categories of coded flexibilities with examples.

Summary of Rule Proposed	Coded Category	Final Code	Example Comment
2017- Allow schools to implement flexibilities regarding milk, sodium, and whole grain requirements during the 2018–2019 school year	Implement newly proposed flexibilities	In favor of flexibilities	“As a registered dietitian nutritionist working in school nutrition, I support this final rule for flexibility with milk, whole grains and sodium requirements.”
	Maintain current regulations	In favor of original HHFKA	“The proposed changes to school meals are deplorable. At a time when 40% of the nation’s children live in poverty, children depend on nutritious meals for substance. The health of these children depends on their access to school meals more than ever now. Please, please don’t dilute children’s meals.
	Implement some of the new flexibilities	In favor of flexibilities	“I say get rid of the whole grains completely. As far as sodium goes I don’t think this should be lowered any further. It is a nightmare now trying to do menus.”
	Implement a different option (new idea)	Other	“As a parent of four children that went through the Public school system, I am well aware of the quality of the schools’ cafeterias. There is NO quality...The use of large amount of “protein” and milk offered is against all of the present scientific studies that speak of more vegetable protein (beans, etc.) and the drink if water as a beverage to prevent obesity & cavities. A Vegan diet for all of the students would satisfy the dietary laws of all religions.”
	Comment on general nutrition	Other	“Children perform better and are healthier over all when fed meals that are abundant in vitamins, minerals, and nutrients. The body will not function properly without those things.”
	Other	Other	“If you were unable to afford the cost of food for your child’s lunch, wouldn’t be nice to know he/she would be able to eat? Without the public embarrassment of going hungry.”
2020- Make implemented flexibilities permanent	Do not restore flexibilities	In favor of original HHFKA	“I object to this change. The children’s nutrition standards should follow Dietary Guidelines for Americans. Lower income families already eat less fiber, more sodium and more saturated fats due to the higher cost of healthy foods. There is no solid foundation in this proposed rule. I urge you to retain standards and not change them.”
	Restore all flexibilities	In favor of flexibilities	“I think this is a great idea in many ways. First, allowing a choice of flavored low-fat milk opposed to plain milk might increase the chance of the student consuming the low-fat milk. Second, allowing the menus to be whole-grain rich offers the student exposure to this healthier alternative of whole grain they might not otherwise get at home. Finally, gradually reducing sodium is a much better plan than eliminating the given amount all at once.
	Restore some flexibilities	In favor of flexibilities	“The flavor milk for students would go over big if a Strawberry flavor is added to the serving for students. Sodium isn’t too much of a problem for seasoning is ok in the planning of meals in our cafeteria. The wheat grains has always been a problem when it comes to serving them to students.”
	Implement a new option (new idea)	Other	“We support the flexibility to offer low fat flavored milk, we would like to offer it once a week for variety in the meals. We do not want to offer it on a daily basis.”
	Comment on general nutrition	Other	“Nutrition Standards should be at the highest level for all of our school age children. Any reduction of standards has a negative impact on the long term health of children.”

**Table 2 nutrients-17-00839-t002:** Federal Register comment breakdown by organization type.

Organization Type	2017 N = 7523 N (%)	2020 N = 2918 N (%)
Constituent, no organization specified	7382 (98.1%)	2586 (88.6%)
Advocacy	60 (0.8%)	65 (2.2%)
School	35 (0.5%)	207 (7.1%)
Academic Researchers	18 (0.2%)	24 (0.8%)
Manufacturer/Food Industry	14 (0.2%)	21 (0.7%)
Other Government Agency—(non-school)	8 (0.1%)	11 (0.4%)
Other	6 (0.1%)	4 (0.1%)

## Data Availability

The original contributions presented in this study are included in the article. Further inquiries can be directed to the corresponding author.

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
