# Peer review of "“Salt and Eat It or No Salt and Trash It?” Shifts in Support for School Meal Program Flexibilities in Public Comments"

_nutrients, 2025, doi:10.3390/nu17050839_

Round 1
Reviewer 1 Report (Previous Reviewer 1)
Comments and Suggestions for Authors
Dear Authors,
running school feeding programs for children is an extremely important challenge (and even an obligation) for the authorities of any country for a number of reasons. Two of the most important are ensuring a healthy start to adult life and reducing social disparities. Therefore, the research issue presented in the manuscript is of high importance.
The manuscript is interesting, the methodology and results are described comprehensively, the Discussion chapter as well. However, the content of this chapter de facto is a summary of the study, and therefore I suggest changing the chapter title. In L. 258, Figure 2 and Figure 3 are announced, but there is no Figure 3 in the manuscript. And there should be, especially since there is a reference to this figure in the Discussion (L. 324 and 325).
The excerpt from the Discussion in lines L. 368-378 is a repetition - but in a broader scope - of the information contained in the Introduction. It does not directly relate to the results of the study, so I suggest removing it from this section and compiling it with the text in the Introduction. Sufficient to conclude the Discussion are the sentences in L. 379-383.
The Conclusion chapter does not refer to the results of the study. Instead, it provides a justification for addressing the research topic, and therefore I suggest moving this section to the Introduction. Besides, the editorial requirements indicate that the Conclusion chapter is optional, so there is no requirement to include it in the manuscript.
And some more suggestions for improving the Introduction chapter. It lacks an important summary/comparison of the recommendations on milk, whole grains and salt in the original documents (HHFKA and Nutrition Standards in the NSL&SBP) and the changes to those recommendations made by amendments to those legislations.
L. 30 - According to WHO, WOF and national health authorities, overweight (or pre-obesity status) is defined as BMI 25-29.9 kg/m², while BMI ≥ 30 kg/m² defines obesity. Can you please explain or reinforce your categorization that assumes BMI>25 is obesity and BMI>30 is severe obesity? L. 44 - NSLP/SBP abbreviations should appear earlier, in L. 36, next to the full names of the programs.
Some portions of the material are highlighted in yellow - this probably means nothing, and the color was left in by mistake.
Kind regards
KR
Author Response
Comment: the content of this chapter de facto is a summary of the study, and therefore I suggest changing the chapter title.
Response: Thank you for your comment. However, I am a little unclear about what you are asking. We changed the title to Shifts in Support for School Meals instead of Analyzing support for school lunch.
Comment: In L. 258, Figure 2 and Figure 3 are announced, but there is no Figure 3 in the manuscript. And there should be, especially since there is a reference to this figure in the Discussion (L. 324 and 325).
Response: Thank you for pointing this out. This was an oversight on our part. We submitted the figure as a supplemental file but not insert it in the paper. It has been added to the manuscript.
Comment: The excerpt from the Discussion in lines L. 368-378 is a repetition - but in a broader scope - of the information contained in the Introduction. It does not directly relate to the results of the study, so I suggest removing it from this section and compiling it with the text in the Introduction. Sufficient to conclude the Discussion are the sentences in L. 379-383.
Response: We have removed this section from the discussion.
Comment: The Conclusion chapter does not refer to the results of the study. Instead, it provides a justification for addressing the research topic, and therefore I suggest moving this section to the Introduction. Besides, the editorial requirements indicate that the Conclusion chapter is optional, so there is no requirement to include it in the manuscript.
Response: The conclusion has been removed.
Comment: And some more suggestions for improving the Introduction chapter. It lacks an important summary/comparison of the recommendations on milk, whole grains and salt in the original documents (HHFKA and Nutrition Standards in the NSL&SBP) and the changes to those recommendations made by amendments to those legislations.
Response: We have added specific information in L66-69 regarding what the HHFKA required in regards to milk, grains, and sodium: “Specifically, schools were required to gradually (over 10 years) reduce sodium levels in school meals, offer only fat‐free plain and flavored milk and low‐fat plain milk, and 100‐percent of grains offered had to be whole grain‐rich [7]. “ L83-87 already explain the changes that the flexibilities allowed “This rule allowed schools flexibility in implementing milk, sodium, and whole grain nutritional requirements during the 2018-2019 school year (SY 2018-2019). Specifically, schools were again allowed to offer flavored, low-fat milk; provide only half of the weekly grains to be whole grain-rich; and retain higher sodium levels.“
Comment: L. 30 - According to WHO, WOF and national health authorities, overweight (or pre-obesity status) is defined as BMI 25-29.9 kg/m², while BMI ≥ 30 kg/m² defines obesity. Can you please explain or reinforce your categorization that assumes BMI>25 is obesity and BMI>30 is severe obesity?
Response: According to the WHO website, “overweight is a BMI greater than or equal to 25; and obesity is a BMI greater than or equal to 30, ” which is the same categorization that the Ogden article that is cited uses. https://www.who.int/news-room/fact-sheets/detail/obesity-and-overweight
Comment: L. 44 - NSLP/SBP abbreviations should appear earlier, in L. 36, next to the full names of the programs.
Response: Thank you for catching this omission. The abbreviations have been added.
Comment: Some portions of the material are highlighted in yellow - this probably means nothing, and the color was left in by mistake.
Response: These are changes that were made to an original submission of the paper and I was asked to highlight the changes that we made. The highlight has been removed from these sections.
Reviewer 2 Report (New Reviewer)
Comments and Suggestions for Authors
The topic is public health nutrition, which is included in the scope of this journal. The whole manuscript is well-written. I suggest improving the quality of the figures. The labels of the graphs are so close that makes it difficult to read them quickly. The conclusion can be improved by considering the nutritional value of salt or not salt in the lunch program.
Author Response
Comment: I suggest improving the quality of the figures. The labels of the graphs are so close that makes it difficult to read them quickly.
Response: Thank you for your suggestion, we have spaced out the labels for Figures 2 and 3 to make them more readable.
Comment: The conclusion can be improved by considering the nutritional value of salt or not salt in the lunch program.
Response: We deleted the conclusion based on recommendations from other reviewers
Reviewer 3 Report (New Reviewer)
Comments and Suggestions for Authors
Regulations regarding collective nutrition for children and adolescents are an important issue in preventing obesity and diseases related to improper nutrition. Analysis of opinions submitted to the Federal Register is to answer the question of whether the changes introduced were met with support or lack thereof by individuals and organizations. The article is an interesting reference to legal changes taking place not only in the USA but also in other countries, including the European Union. The authors have started a discussion on an important topic that may be enriched with the positions of other organizations dealing with collective nutrition for children.
The study is well designed and unbiased.
The references in the bibliography are appropriate.
Suggested minor corrections:
1. It is worth paying attention to the yellow markings, which may be related to the previous review?
2. It is worth moving figure 2 lower so that it does not divide lines 277 and 278 in this way.
3. There are additional descriptions under Figures, which either need to be entered into the main text or described so that they do not repeat the title of the figure placed above.
Author Response
Comment: It is worth paying attention to the yellow markings, which may be related to the previous review?
Response: These are changes that were made to an original submission of the paper and I was asked to highlight the changes that we made. The highlight has been removed from these sections.
Comment: It is worth moving figure 2 lower so that it does not divide lines 277 and 278 in this way.
Response: This will likely change in final layout, but will work with editors to makes sure it does not divide lines.
Comment: There are additional descriptions under Figures, which either need to be entered into the main text or described so that they do not repeat the title of the figure placed above.
Response: The journal guidelines for figures ask for a description to accompany the figure as a possible stand alone.
Reviewer 4 Report (New Reviewer)
Comments and Suggestions for Authors
1. Abstract & Introduction:
Lack of a clear hypothesis or research question
Background information is partially insufficient
2. Methods:
Unclear details on sample selection
No justification for certain methodological choices
Lack of discussion on potential biases or limitations
3. Results:
Insufficient statistical analysis
Lack of transparency in data presentation
No consideration of confounders
4. Discussion:
Insufficient critical engagement with own findings
Limited reference to relevant literature
Overinterpretation of results
5. Conclusion:
No clear link to the original research question
Overly broad or insufficiently justified claims
Author Response
Comment: Abstract & Introduction: Lack of a clear hypothesis or research question
Response: Please see additions in L101-105. We have added research questions into the introduction “Our research questions include:
(1) Was the predominant argument in favor or in opposition of the flexibilities?
(2) How did support vary among different stakeholder groups?
(3) What were the main topics used in comments in support and in opposition of flexibilities? “
Comment: Background information is partially insufficient
Response: We added information regarding the initial implementation of Health Hunger Free Kids Act and additional information. See added text in L63-81
Comment: Unclear details on sample selection
Response: The sample consists of comments submitted to the Federal Register. The process to extract these and the amount extracted are described in lines 147-156 and 209-2013.
Comment: No justification for certain methodological choices
Response: We have expanded the justification on the use BERTopic in the manuscript:
“we use topic modeling to identify recurring topics behind these massive amounts of tex-tual data… … We choose BERTopic because it leverages pre-trained language models, which allows it to better semantically contextualize these comments and generate more meaningful topics compared to traditional topic models like LDA.”
Comment: Lack of discussion on potential biases or limitations
Response: We have acknowledged additional biases and limitations in the manuscript that were previously not extensively discussed. These include:
- The possibility that some districts may actively encourage parents or stakeholders to engage with the U.S. Federal Register. As a result, comments from these districts may be overrepresented, potentially leading to an imbalance in the dataset. Consequently, methods such as BERTopic may amplify these discrepancies in representation.
- Consequently, stakeholders such as parents or educators from older demographics may have lower levels of online engagement, leading to their underrepresentation such datasets collected via digital platforms.
- Active individuals on the Federal Register who are less directly affected by the policy may be influenced by previously submitted comments, leading them to echo popular sentiments that resonate with them. This could result in the over-amplification of popular topics reflected in our topic models.
Comment: Insufficient statistical analysis
Response: Our research questions are not appropriate for hypothesis testing.
Comment: Lack of transparency in data presentation
Response: We have added details in Figure 2 and 3, including:
- Expanding on the caption to detail the x-axis:
“… The x-axis represents the topic models across each organization type…”
- Expanding the caption to include a description on how the y-axis – “normalized document frequency” is formulated:
“… The y-axis represents the number of comments from an organization that have been assigned to a specific topic, normalized by the total number of comments from that organization …”
Comment: No consideration of confounders
Response: The topic modeling analysis was designed to complement the stances that stemmed from coding, providing a more subtle understanding surrounding the distinct perspectives across stances across organizations, thereby allowing us to understand the confounding factors surrounding the stances took across stakeholders.
Comment: Insufficient critical engagement with own findings, Limited reference to relevant literature, Overinterpretation of results
Response: I am unsure what statements you are referring to or how we are not engaging with our findings. I am also unsure what other literature we are not recognizing in our writing.
Comment: Conclusion: No clear link to the original research question, Overly broad or insufficiently justified claims
Response: We have deleted the conclusion.
Round 2
Reviewer 4 Report (New Reviewer)
Comments and Suggestions for Authors
None
Comments on the Quality of English Language
Accept
Author Response
Comment: 1st paragraph (lines 28-31):
Obesity among people aged 2-19 in the United States is a profound issue that effects an increasing number of youth.[1] Specifically, for this age range, obesity (body mass in- dex of >25) has increased from 13.9% from 2000 to 19.3% in 2018, and severe obesity (body mass index of >30) increased from 3.6% to 6.1%.[1]
The response does not adequately address the reviewer’s concern, as the BMI cut-off points mentioned (BMI ≥ 25 for overweight and BMI ≥ 30 for obesity) apply to adults. In children and adolescents, classification is based on specific percentiles from WHO or CDC growth charts. I recommend that the author refer to the appropriate guidelines for defining obesity in youth and revise the wording to avoid misclassification Reply: I apologize for this oversight. We have corrected the information to be specific to children and revised accordingly in L30-33:
Specifically, for this age range, obesity (body mass index values at or above the 95th percentile of the growth charts) a) has increased from 13.9% from 2000 to 19.3% in 2018, and severe obesity (body mass index at or above 120% of the 95th percentile) increased from 3.6% to 6.1%.
This manuscript is a resubmission of an earlier submission. The following is a list of the peer review reports and author responses from that submission.
Round 1
Reviewer 1 Report
Comments and Suggestions for Authors
Dear Authors,
The issue of healthy eating for children and getting them off to a healthy start is extremely important for many reasons, including the growing prevalence trend of overweight and obesity in this population group. It is also extremely important in an American society that is multiracial and has large disparities in household budgets. From this point of view, the purpose of the conducted analysis of comments posted in the Federal Register was a good idea. I expected that the results of the analysis could be used in the creation of a federal school nutrition policy.
Meanwhile, I have some important concerns and suggestions.
In the Introduction, it would have been helpful to outline the USDA's policy on school meals for children more broadly. A thorough description of the NSLP/SBP is missing. It is not clear whether all schools mandatorily participate in this programme, what is the ratio of financial and food support from the USDA, what percentage of students are covered etc. The assumptions/ provisions of the HHFKA are also not written about.
Table 1 should be included in the Results section. Firstly because it includes the code categories and final codes in the text of subsection 2.2. Secondly, it contains quotes of comments and, in combination with columns 2 and 3, shows the results of the analysis.
What is missing from the analysis of the results is a quantitative and qualitative presentation of the comments on milk, whole grains, sodium, and this would certainly increase the scientific value of the article.
In Chapter 4 Discussion, the first and second paragraphs are de facto the results from the study. Only the third paragraph can possibly be considered as part of the discussion. This means that there is no discussion of the results in the manuscript and it would be appropriate to have one chapter Results and Discussion.
In Study Limitations it is not worth writing about the lack of representativeness of the comments, as this is obvious in the public consultation. Also the limitation is not that "sample sizes in 2017 and 2020 could have different organisation compositions and overall sizes", because in the methodology the authors explained how they dealt with this.
Conclusions do not follow from the research, the entire first paragraph could have been part of a discussion of the results. The authors also did not provide an idea of what to do next with these results - whether they will resonate in the public debate of children's equality in access to healthy nutrition.
Last, but not least, the manuscript is written carelessly, with many punctuation marks missing or too many, no inverted commas in section 3.2. especially where there are quotations of statements (what is a "Milk-Our district"? - L.174 ), full stops at the end of sentences are before, not after, the reference number, there are spelling errors (E.g. L. 9 - there is an A and a The; L. 11 - not Stated, but States; L. 270/271 - presumably, the Authors had in mind that commenters may believe that the sending of a comment will NOT have an impact on the policy decision-making by the federal agency), the Figure 1 legend is missing, etc.
Comments on the Quality of English Language
The manuscript is written carelessly, with many punctuation marks missing or too many, no inverted commas in section 3.2. especially where there are quotations of statements (what is a "Milk-Our district"? - L.174 ), full stops at the end of sentences are before, not after, the reference number, there are spelling errors (E.g. L. 9 - there is an A and a The; L. 11 - not Stated, but States; L. 270/271 - presumably, the Authors had in mind that commenters may believe that the sending of a comment will NOT have an impact on the policy decision-making by the federal agency), the Figure 1 legend is missing, etc.
I am sorry, but the peer-reviewed manuscript represents low scientific quality.
Author Response
Thank you for your thorough and thoughtful review of our manuscript. We have addressed the revisions you have suggested as outlined below:
Comment: In the Introduction, it would have been helpful to outline the USDA's policy on school meals for children more broadly. A thorough description of the NSLP/SBP is missing. It is not clear whether all schools mandatorily participate in this programme, what is the ratio of financial and food support from the USDA, what percentage of students are covered etc. The assumptions/ provisions of the HHFKA are also not written about.
Response: Thank you for this suggestion. We have added addition information regarding the school lunch and breakfast programs in lines 43-48 in the Introduction.
Comment: Table 1 should be included in the Results section. Firstly because it includes the code categories and final codes in the text of subsection 2.2. Secondly, it contains quotes of comments and, in combination with columns 2 and 3, shows the results of the analysis.
Response: Thank you for this suggestion. We moved Table 1 to the Results section.
Comment: What is missing from the analysis of the results is a quantitative and qualitative presentation of the comments on milk, whole grains, sodium, and this would certainly increase the scientific value of the article.
Response: We were able to complete some qualitative analyses on the comments regarding comments supporting specific types of flexibilities. This has been added into the results section (L.245-262). Table 3 has also been added.
Comment: In Chapter 4 Discussion, the first and second paragraphs are de facto the results from the study. Only the third paragraph can possibly be considered as part of the discussion. This means that there is no discussion of the results in the manuscript and it would be appropriate to have one chapter Results and Discussion.
Response: We have removed the conclusion section and added to the discussion section to make it more of a discussion about the results.
Comment: In Study Limitations it is not worth writing about the lack of representativeness of the comments, as this is obvious in the public consultation. Also the limitation is not that "sample sizes in 2017 and 2020 could have different organisation compositions and overall sizes", because in the methodology the authors explained how they dealt with this.
Response: Thank you for your suggested revision. We have removed the sentence regarding the sample sizes.
Comment: Conclusions do not follow from the research, the entire first paragraph could have been part of a discussion of the results. The authors also did not provide an idea of what to do next with these results - whether they will resonate in the public debate of children's equality in access to healthy nutrition.
Response: We appreciate this suggestion. We have moved components of the conclusion into to the discussion section
Comment: Last, but not least, the manuscript is written carelessly, with many punctuation marks missing or too many, no inverted commas in section 3.2. especially where there are quotations of statements (what is a "Milk-Our district"? - L.174 ), full stops at the end of sentences are before, not after, the reference number, there are spelling errors (E.g. L. 9 - there is an A and a The; L. 11 - not Stated, but States; L. 270/271 - presumably, the Authors had in mind that commenters may believe that the sending of a comment will NOT have an impact on the policy decision-making by the federal agency), the Figure 1 legend is missing, etc
Response: Thank you for catching these grammatical errors. We have made the corrections.
Reviewer 2 Report
Comments and Suggestions for Authors
The authors have chosen a very relevant and important topic, using qualitative methodology to examine public opinions related to the School Lunch Program Flexibilities. The manuscript is well-structured and logically organized. I suggest considering the following revisions:
I recommend expanding the methodological description with additional information on coding, such as the number of coders for the comments, the results of interrater reliability, and whether any comments were excluded from the analysis.
In the results section, it would be beneficial to mark verbatim quotes from the comments with quotation marks for easier readability.
I suggest rephrasing the conclusion section to summarize the most important findings and conclusions.
Author Response
Thank you for your thorough and thoughtful review of our manuscript. We have addresses your suggested revisions as outlined below:
Comment: I recommend expanding the methodological description with additional information on coding, such as the number of coders for the comments, the results of interrater reliability, and whether any comments were excluded from the analysis.
Response: We apologize for the oversight. We have added in more information regarding coding and the inter-rater reliability in lines 126-129
Comment:In the results section, it would be beneficial to mark verbatim quotes from the comments with quotation marks for easier readability.
Response: We have added in quotation marks
Comment: I suggest rephrasing the conclusion section to summarize the most important findings and conclusions.
Response: Thank you for your suggestion. We have revised the discussion and conclusion